# Discovery of New Glucose Uptake Inhibitors as Potential Anticancer Agents by Non-Radioactive Cell-Based Assays

**DOI:** 10.3390/molecules27228106

**Published:** 2022-11-21

**Authors:** Hsueh-Chih Hung, Li-Cheng Li, Jih-Hwa Guh, Fan-Lu Kung, Lih-Ching Hsu

**Affiliations:** School of Pharmacy, College of Medicine, National Taiwan University, Taipei 10617, Taiwan

**Keywords:** GLUT1 inhibitors, 2-NBDG, glucose uptake assays, anticancer, breast and ovarian cancer

## Abstract

Tumor cells rely on aerobic glycolysis to support growth and survival, thus require more glucose supply. Glucose transporters GLUTs, primarily GLUT1, are overexpressed in various cancers. Targeting GLUTs has been regarded as a promising anticancer strategy. In this study, we first evaluated 75 potential GLUT1 inhibitors obtained from virtual screening of the NCI chemical library by a high-throughput cell-based method using a fluorescent glucose analogue 2-(*N*-(7-nitrobenz-2-oxa-1,3-diazol-4-yl)amino)-2-deoxy-d-glucose (2-NBDG) in COS-7 and SKOV3 cells that express high levels of GLUT1. Four compounds, #12, #16, #43 and #69, that significantly inhibited glucose uptake were further evaluated using flow cytometry directly measuring 2-NBDG uptake at the single-cell level and a Glucose Uptake-Glo^TM^ assay indirectly measuring 2-deoxy-d-glucose uptake in SKOV3, COS-7 or MCF-7 cells. The inhibitory effect on cancer cell growth was also determined in SKOV3 and MCF-7 cells, and #12 exhibited the best growth inhibitory effect equivalent to a known GLUT1 inhibitor WZB117. Although the anticancer effect of the identified potential GLUT1 inhibitors was moderate, they may enhance the activity of other anticancer drugs. Indeed, we found that #12 synergistically enhanced the anticancer activity of metformin in SKOV3 ovarian cancer cells.

## 1. Introduction

Unlike normal cells, cancer cells rely more on glycolysis for energy production, even in an environment with an abundant oxygen supply, a phenomenon called the Warburg effect [1]. Studies have indicated that mitochondrial defects and malfunction, adaptation to hypoxic tumor microenvironment, oncogenic signaling and abnormally altered metabolic pathways may be the underlying mechanisms of the Warburg effect, and targeting glycolysis is considered a good strategy to selectively kill cancer cells [2]. Glycolysis produces much less ATP than oxidative phosphorylation. Thus, tumor cells take up more glucose than most normal cells and this phenomenon has become the basis of visualizing cancer cells by imaging, for example, using [^18^F] fluoro-2-deoxyglucose positron emission tomography [3]. Since cancer cells are addicted to glucose, they may be more sensitive to glucose deprivation-induced growth inhibition or cell death. Human glucose transporters, including active sodium-coupled glucose cotransporters (SGLTs), and passive facilitative glucose transporters (GLUTs), play an important role in cellular glucose uptake across the plasma membrane. GLUTs belong to a family of transmembrane proteins with 14 members divided into three classes based on sequence similarity. Class I GLUTs include GLUT1-4 and GLUT14, class II GLUTs include GLUT5, GLUT7, GLUT9 and GLUT11 and class III GLUTs include GLUT6, GLUT8, GLUT10, GLUT12 and HMIT. These GLUT isoforms have different substrate specificities, transport capabilities and tissue distribution, and play important roles regulating glucose uptake and metabolism. Among them, GLUT1-4 have been well studied. Normally, GLUT1 is expressed in erythrocytes and the blood brain barrier of the brain, and is responsible for basal glucose uptake. GLUT2 is mainly expressed in the liver, small intestine, kidney and pancreas for transport of glucose and fructose, and glucose sensing. GLUT3 is expressed in neurons. GLUT4 is expressed in muscle, heart and adipose tissue, and is responsible for insulin-stimulated glucose transport. Tumor cells usually overexpress GLUTs, primarily GLUT1; therefore, GLUT1 could be a potential target for anticancer therapy [4,5].

Several natural products, such as phloretin, resveratrol and cytochalasin B have been identified as GLUT1 inhibitors [5]. In recent years, many GLUT1 inhibitors have also been synthesized and demonstrated to inhibit cancer cell growth [6]. For example, WZB117 was identified as a GLUT1 inhibitor with a good activity against non-small cell lung cancer cells both in vitro and in vivo [7]. Ung and co-workers performed a virtual screening of commercial libraries based on the X-ray structure of the *E. coli* xylose transporter and discovered PUG-1 and PUG-7 as new GLUT1 inhibitors [8]. Furthermore, the discovery of a highly selective GLUT1 inhibitor BAY-876 was also reported [9].

Methods for the identification of GLUT inhibitors include direct measurement of the uptake of radioisotope-labeled 2-deoxy-d-glucose (2DG), as in the identification of WZB117 [7], or fluorescent analogues of d-glucose, such as 2-(*N*-(7-nitrobenz-2-oxa-1,3-diazol-4-yl) amino)-2-deoxy-d-glucose (2-NBDG) [10,11]. In addition, fluorescence [12] or luminescence [13] detection methods are also available for indirect measurement of 2DG uptake.

2-NBDG, synthesized by Yoshioka et al. in 1996, is a fluorescent analogue that has Km values comparable to d-glucose. Further studies have indicated that 2-NBDG is metabolized to 2-NBDG 6-phosphate and then decomposes to a non-fluorescent derivative inside the *E. coli* cell [10,14]. 2-NBDG is useful for monitoring glucose uptake in mammalian cells via direct visualization. The uptake of 2-NBDG can also be detected at the single-cell level by flow cytometry or applied to the development of high-throughput quantitative assays to allow large-scale measurement of glucose uptake in cancer cells [10,11,15,16].

Interestingly, many GLUT1 inhibitors have been identified serendipitously but not from direct screening for GLUT1 inhibitors. The crystal structure of the human GLUT1 (hGLUT1) in an inward-open (IOP) conformation has been solved [17]; this would facilitate in silico screening of GLUT1 inhibitors. Molecular docking studies have also revealed that the IOP is the preferred conformation for ligand binding to hGLUT1 [18]. We have synthesized a novel fluorescent glucose analogue 1-NBDG and used it to establish a non-radioactive cell-based method for the screening of SGLT1 and SGLT2 inhibitors [19,20]. Here we used a similar approach to set up a non-radioactive cell-based method using 2-NBDG in COS-7 monkey kidney cells, as well as SKOV3 human ovarian cancer cells that overexpress GLUT1, to evaluate and validate 75 potential GLUT1 inhibitors. These were obtained from virtual screening of the National Cancer Institute (NCI) chemical library, which contains over 140,000 compounds, based on the crystal structure of hGLUT1 [17]. Four best hits were further evaluated using other non-radioactive cell-based glucose uptake assays in COS-7 cells, SKOV3 cells and MCF-7 human breast cancer cells. We found that different assay methods and cell lines used for the analysis led to slightly different results, and the discrepancy of different screening methods was compared and discussed. We also showed that although the anticancer effect of identified GLUT1 inhibitors was moderate, they may enhance the activity of other anticancer agents.

## 2. Results

### 2.1. Screening of Potential GLUT1 Inhibitors by High-Throughput 2-NBDG Uptake Assay

Radioisotope-labeled 2DG, such as ^3^H-2DG, is frequently used to measure glucose uptake, but the use of radioisotopes is costly and causes safety concerns. A more cost effective and safer non-radioactive cell-based method using 2-NBDG was then set up for the screening of novel GLUT1 inhibitors in a 96-well format. COS-7 and SKOV3 cells that express high levels of GLUT1 were chosen for the high-throughput screening (HTS) method. Cells may express other GLUT isoforms, such as GLUT2. Reverse transcription-polymerase chain reaction (RT-PCR) results indicated that GLUT1 but not GLUT2 was expressed in COS-7 and SKOV3 cells; however, both GLUT1 and GLUT2 were detected in HepG2 human hepatocellular carcinoma cells (Figure 1A). It remains to be determined whether COS-7 and SKOV3 cells express other GLUT isoforms. Nevertheless, it has been reported that GLUT1 but not GLUT2, 3 or 4 is overexpressed in human ovarian cancer [21]. Thus, compounds that can inhibit glucose uptake in COS-7 and SKOV3 could be potential GLUT1 inhibitors.

Conditions including cell number, incubation time and 2-NBDG concentration were optimized in COS-7 cells, and phloretin was used as an inhibitor of glucose uptake. As shown in Figure 1B, phloretin significantly inhibited 2-NBDG uptake after 60 and 90 min of incubation, and incubation for 90 min provided a better detection window for the screening of GLUT1 inhibitors. However, the maximal inhibition of glucose uptake measured by this method was approximately 50%. Thus, this HTS method was employed as an initial screening, and other methods were subsequently used for further validation of hits.

Seventy-five potential GLUT1 inhibitors from the NCI chemical library were tested by the high-throughput 2-NBDG uptake assay in COS-7 cells at 100 μM except otherwise indicated. Phloretin at 50 μM was routinely used as a positive control since not much difference was observed between 50 μM (56.38 ± 3.63% uptake) and 100 μM (53.83 ± 0.87% uptake). The results indicated that #43 had the strongest GLUT inhibitory effect with 59.68 ± 2.28% 2-NBDG uptake (~40% inhibition) in COS-7 cells, while other compounds, including #11, #12, #14, #16, #31, #40, #46, #48 and #69, exhibited 15-25% inhibition of glucose uptake as summarized in Appendix A.

SKOV3 cells were used next for 2-NBDG uptake assay to further verify the top 10 potential GLUT1 inhibitors obtained from the primary screening in COS-7 cells, and #12, #16, #43 and #69 inhibited more than 30% of glucose uptake in SKOV3 cells (Figure 2A). The comparison of 2-NBDG uptake results of these four compounds in COS-7 and SKOV3 cells is shown in Figure 2B. Compound #43 was the most potent compound in COS-7 cells, whereas #12 (56.46 ± 8.05%) was slightly more potent than other compounds, and as potent as 50 μM (57.58 ± 1.78%) (Figure 2B) or 100 μM of phloretin (55.58 ± 2.87%) (Table 1) in SKOV3 cells.

### 2.2. Validation of the Potential GLUT1 Inhibitors by Flow Cytometry Assay

Single cell-based flow cytometry assay of 2-NBDG uptake was also used to confirm the GLUT1 inhibitory effect. Results showed that #12, #16, #43 and #69 all inhibited 2-NBDG uptake in SKOV3 cells as revealed by the shift of the FL1-H peaks toward the left compared to the vehicle control (Figure 3A).

MCF-7 cells also overexpressed GLUT1 as indicated in Figure 1A. However, MCF-7 cells were not suitable for the population-based high-throughput 2-NBDG uptake assay described above due to a tendency to detach during the assay procedure. Therefore, 2-NBDG uptake assay by flow cytometry was conducted to measure the inhibitory effect of #12, #16, #43 and #69. Representative histograms are shown in Figure 3B with geomean of each sample indicated. Quantitative analysis of 2-NBDG uptake calculated from the geomean revealed that #43 was most potent and its activity was comparable to the known GLUT1 inhibitors phloretin and WZB117 (Figure 3C and Table 1). In addition, #12, #16 and #69 also significantly inhibited 20–40% of 2-NBDG uptake in MCF-7 cells.

### 2.3. Validation of the Potential GLUT1 Inhibitors by Glucose Uptake-Glo™ Assay

To further validate the potential GLUT1 inhibitors selected by the primary screening, we also used a commercial kit, the Glucose Uptake-Glo™ assay, to indirectly measure 2DG uptake by luminescence [13]. Cells were seeded in 96-wells and pretreated with compounds for 10 min at 37 °C before 2DG was added for further incubation for 10 min at room temperature. 2DG is converted to 2DG 6-phosphate (2DG6P) by hexokinase once transported into the cell. 2DG6P, not metabolized further and trapped inside the cell, can be quantified by coupling to an enzymatic detection system to generate luminescence. The detection reagent contains glucose 6-phosphate dehydrogenase (G6PDH), NADP^+^, reductase, luciferase and proluciferin. 2DG6P can be oxidized to 6-phosphodeoxygluconate, and NADP^+^ is simultaneously reduced to NADPH by G6PDH. NADPH is then used by reductase to convert proluciferin to luciferin which is used by luciferase to generate a luminescent signal that is proportional to the concentration of 2DG6P. The results of 2DG uptake in COS-7 cells treated with 100 μM of #12 (86.04 ± 1.06%) and #16 (71.27 ± 1.40%) obtained from the Glucose Uptake-Glo™ assay were comparable to those obtained from the 2-NBDG uptake assay (77.46 ± 3.82% and 74.36 ± 2.99%, respectively). In contrast, 2DG uptake in COS-7 cells was dramatically inhibited to 6.38 ± 2.36% by 100 μM #43 and 15.24 ± 3.46% by 100 μM #69. Thus, the activity of #43 and #69 measured by the Glucose Uptake-Glo™ assay was much more potent compared to the 2-NBDG uptake assay (59.68 ± 2.28% and 76.96 ± 5.91% uptake, respectively). Nevertheless, 100 μM #43 inhibited 2DG and 2-NBDG uptake as effectively as 100 μM phloretin with 8.72 ± 3.22% 2DG uptake and 53.83 ± 0.87% 2-NBDG uptake in COS-7 cells (Figure 4A and Table 1).

As described above, 2-NBDG uptake results obtained from different cell lines varied to a certain extent (Figure 2 and Figure 3). Therefore, the Glucose Uptake-Glo™ assay was also performed in SKOV3 and MCF-7 cells. Unlike in COS-7 cells, inhibition of 2DG uptake by #12, #16, #43 and #69 was much less effective in SKOV3 cells. At 100 μM, phloretin significantly suppressed 2DG uptake down to 3.81 ± 1.35%, whereas, the most potent compound #43 inhibited 2DG uptake by ~50% (50.03 ± 3.34% uptake), and #12, #16 and #69 resulted in 88.06 ± 10.40%, 77.11 ± 7.40% and 67.96 ± 7.82% 2DG uptake, respectively, in SKOV3 cells (Figure 4A and Table 1).

In MCF-7 cells, only #43 exhibited a significant inhibitory effect on 2DG uptake (29.24 ± 2.52% uptake), while #12, 16 and 69 did not show any inhibitory activity at 50 μM. Both 50 μM of phloretin (10.32 ± 2.76%) and WZB117 (2.43 ± 0.25%) markedly inhibited 2DG uptake in MCF-7 cells in the Glucose Uptake-Glo™ assay (Figure 4B and Table 1).

### 2.4. Comparison of Different Glucose Uptake Assay Methods in COS-7, SKOV3 and MCF-7 Cells

Many GLUT inhibitors have been identified by researchers using a variety of cell-based assays including direct or indirect measurements of glucose uptake [5,6]; however, it is not clear whether it is appropriate to directly compare data reported by different groups obtained using different methods and cell lines. In this study, we employed several glucose uptake methods to evaluate and identify novel GLUT1 inhibitors in different cell lines, and known GLUT1 inhibitors phloretin and WZB117 were included for comparison in some assays. Comparison of glucose uptake evaluated directly by 2-NBDG uptake, either population-based HTS (indicated as 2-NBDG) or single cell-based flow cytometry (indicated as 2-NBDG flow), and indirectly by 2DG uptake (indicated as 2DG) is summarized in Table 1 from data illustrated in Appendix A, and Figure 2, Figure 3 and Figure 4. The maximal inhibition of glucose uptake measured by HTS 2-NBDG uptake was limited to 40-50%, while when measured by 2-NBDG flow could reach ~70% inhibition, reflecting the discrepancy between population and single cell-based 2-NBDG uptake assay. 2DG uptake assay provided a wider detection window for the screening of GLUT inhibitors. Phloretin at 100 μM resulted in 53.83% 2-NBDG uptake vs. 8.72% 2DG uptake, and 100 μM #43 led to 59.68% 2-NBDG uptake vs. 6.38% 2DG uptake in COS-7 cells. In SKOV3 cells, 100 μM phloretin resulted in 55.58% 2-NBDG uptake vs. 3.81% 2DG uptake, and 100 μM #43 led to 67.66% 2-NBDG uptake vs. 50.03% 2DG uptake. Interestingly, 100 μM #12 had the best inhibitory effect with 56.46% 2-NBDG uptake, but led to 88.06% 2DG uptake in SKOV3 cells. In MCF-7 cells, 50 μM phloretin resulted in 36.87% 2-NBDG uptake by flow cytometry vs. 10.32% 2DG uptake, and 50 μM #43 led to 35.84% 2-NBDG uptake by flow cytometry vs. 29.24% 2DG uptake. These discrepancies may be in part due to differences in the detection of glucose uptake (direct vs. indirect detection), incubation time (30–90 min vs. 10 min), concentrations of substrates used (200 μM 2-NBDG vs. 1 mM 2DG) or GLUT1 expression levels. Further investigation is required to clarify this issue. WZB117 was most potent among all compounds tested in MCF-7 cells regardless of using 2-NBDG or 2DG assay, resulting in 27.35% 2-NBDG uptake by flow cytometry or 2.43% 2DG uptake at 50 μM (Table 1).

### 2.5. Effects of Potential GLUT1 Inhibitors on the Growth of SKOV3 and MCF-7 Cells

The chemical structures of #12 (NSC111720), #16 (NSC126757), #43 (NSC36525) and #69 (NSC319424) are shown in Figure 5A. They possess aromatic or non-aromatic nitrogen-containing heterocyclic structures commonly found in highly potent GLUT inhibitors [6]. To our knowledge, they have not been reported as GLUT1 inhibitors. The chemical structures of known GLUT1 inhibitors phloretin, resveratrol, cytochalasin B, WZB117 and BAY-876 are provided in Appendix A for comparison. The effects of these four compounds on cell proliferation were determined and compared with phloretin and WZB117. At 100 μM, #12 had the best inhibitory effect (31.59 ± 2.99% cell viability) and WZB117 was less potent (57.74 ± 3.68%), while #16, #43 and #69 showed very moderate effects (74.0–76.5%), and phloretin had the smallest effect (89.82 ± 3.49%) on SKOV3 cell growth (Figure 5B). In MCF-7 cells, #12 (25.5 ± 3.23% cell viability) and WZB117 (26.54 ± 3.64%) were equally potent, while phloretin and #69 barely showed any anticancer effect at 100 μM after 72 h of treatment. Although less potent than #12 and WZB117 at 100 μM with a cell viability of 54.18 ± 1.00%, #43 showed the best growth inhibition (69.25 ± 0.91% cell viability) compared to #12 (89.39 ± 3.60%) or WZB117 (81.28 ± 1.47%) at 25 μM (*p* < 0.01, two-tailed *t*-test) in MCF-7 cells (Figure 5C).

### 2.6. The Combination of #12 and Metformin Synergistically Inhibits SKOV3 Cell Growth

Since GLUT1 inhibitors identified in this study only displayed moderate anticancer effects, we sought to explore the possibility of combination treatment with other anticancer drugs. Among the four GLUT1 inhibitors identified, #12 showed the best inhibitory effects on both 2-NBDG uptake and growth in SKOV3 cells; therefore, #12 was chosen for further combination studies.

Previous studies indicated that metformin, a widely used antidiabetic agent, when combined with 2DG, inhibited mitochondrial respiration and glycolysis in prostate cancer cells, leading to a severe depletion of ATP [22]. In light of this successful strategy, the combinatorial effect of #12 and metformin was determined in SKOV3 cells. The molar ratio of #12 and metformin IC_50_ values in SKOV3 cells was approximately 1:200; the 1:50, 1:100 and 1:200 molar ratios were then tested in a preliminary study, and with 1:100 exhibiting the best combinatorial effect, it was chosen for further studies. As shown in Figure 6A, the combination of #12 and metformin at a molar ratio of 1:100 exhibited a strong synergistic effect with a combination index at 50% growth inhibition (CI_50_) value of 0.48. The 50% inhibitory concentration (IC_50_) of #12 alone was 86.2 μM, but dropped to 27.91 μM in combination treatment. Likewise, the IC_50_ of metformin was estimated to be 18.48 mM as a single agent, but dropped dramatically down to 2.79 mM in combination treatment. Interestingly, the combination of WZB117, a known GLUT1 inhibitor, and metformin also had a moderate synergism with a CI_50_ value of 0.78 (Appendix A). The long-term growth inhibitory effect was examined by the colony formation assay, and a low-dose combination (10 μM #12 and 1 mM metformin) significantly suppressed colony formation from 65.36 ± 3.27% and 40.57 ± 4.22% in #12 and metformin single treatments, down to 1.22 ± 0.03% in combination treatment (Figure 6B). The combination of 10 μM WZB117 and 1 mM metformin also showed a similar effect on colony formation (Appendix A).

Apoptosis assay by Annexin V-FITC/propidium iodide (PI) double staining was performed in SKOV3 cells treated with 100 μM #12 and/or 10 mM metformin for 72 h. As shown in Figure 6C, apoptotic cells (early apoptosis and late apoptosis) were significantly increased by the combination treatment (51.77 ± 2.05%) compared to the control (4.64 ± 0.46%), #12 (35.25 ± 4.14%) or metformin (4.08 ± 0.73%) alone, indicating that the combination of #12 and metformin synergistically induced apoptosis in SKOV3 cells.

Recent studies revealed that aerobic glycolysis supplied the primary energy for cancer cell motility and cytoskeleton rearrangement [23,24], and the combination of metformin and 2DG significantly inhibited the migration and invasion of ovarian cancer cells [25]. To determine whether the combination of #12 and metformin affected SKOV3 cell migration, the wound healing assay was performed. Results shown in Figure 6D indicated that the combination of 100 μM #12 and 10 mM metformin significantly inhibited wound healing (17.73 ± 3.42% wound healing in combination) compared to single agents (#12: 41.4 ± 3.23% and metformin: 87.07 ± 1.91%) after 24 h of treatment.

## 3. Discussion

The population-based 2-NBDG uptake assay in the 96-well format was set up and used as an initial method for the validation of hits from in silico screening of GLUT1 inhibitors. This method provided a rapid and direct glucose uptake measurement to determine the actviity of GLUT1 inhibitors. However, there are some limilations of this system. Firstly, it is not suitable for fluorescent compounds since inherent fluorescence of the compounds may interfere with the detection of the inhibitory effect leading to false-negative results, or the compounds may be mistaken as GLUT1 enhancers, as in the case of #19 and #72 (Appendix A). Thus, it is critical to determine whether test compounds possess intrinsic fluorescence. Secondly, if test compounds are highly cytotoxic, the reduced fluorescence signal caused by severe cell loss may lead to false-positive results as in #3, #8 and #25 (Appendix A). Furthermore, it is not suitable for cells that cannot adhere well in microliter plates during the assay process as in the case of MCF-7 cells. Thus, examining cell conditions under the microscope before cell lysis for fluorescence detection is important to avoid false results. Although this system has some shortcomings, it provides a safe and cost-effective method for the primary screening.

Cytochalasin B, a natural product with potent GLUT1 inhibitory activity, and two novel Phe amide-derived GLUT1 inhibitors, GLUT-i1 and GLUT-i2, have been reported to bind in the central cavity of hGLUT1 in the IOP conformation, where Trp388 is the common interacting residue based on the co-crystal structure [26]. Molecular docking using hGLUT1-*β*-NG co-crystal model (PDB ID: 4PYP) [17] revealed that #12, #16, #43, #69 and WZB117 all bound to hGLUT1 in the IOP conformation (Appendix A). Interestingly, both #12 and WZB117 interacted with Gln283 and Trp388 of hGLUT1 via hydrogen bonds (Appendix A).

The correlation between GLUT1 inhibition and anticancer activity of known GLUT1 inhibitors has been controversial. BAY-876, reported as a highly selective and potent GLUT1 inhibitor with an IC_50_ value of 2 nM as determined by indirect measurement [9], did not exhibit any growth inhibitory effect toward triple-negative breast cancer cells at 3 μM [27]. In contrast, BAY-876 effectively inhibited the growth of SKOV3 and OVCAR3 cells with IC_50_ values of 188 nM and ~60 nM, respectively [21]. Liu et al. reported that WZB117, another potent GLUT1 inhibitor which inhibited glucose uptake with an IC_50_ value of 0.5 μM in A549 cells determined by ^3^H-2DG uptake assay, inhibited cancer cell growth with IC_50_ values of ~10 μM in both A549 and MCF-7 cells [7]. However, Chen et al. reported that the IC_50_ of WZB117 in MCF-7 cells was 42.66 μM [28]. Our previous study also showed that the IC_50_ of WZB117 was between 40 and 60 μM in MCF-7 cells [29]. Phloretin was a more potent GLUT1 inhibitor than the four novel compounds identified in this study based on most of the assay methods, however, it barely showed any growth inhibitory effect in the concentration range tested (25–100 μM) in both SKOV3 and MCF-7 cells treated for 72 h followed by the MTT assay. It was reported that IC_50_ values of phloretin ranged between 36-197 μM in four breast cancer and four ovarian cancer cell lines [30], but cells were treated for 5 days and cell growth was determined by the sulforhodamine B assay, which usually gives rise to lower IC_50_ values than those obtained by the MTT assay. For #12 in SKOV3 cells, 2-NBDG (~56% uptake or ~44% inhibition) but not 2DG uptake inhibition (~88% uptake or ~12% inhibition) (Table 1) correlated better with growth inhibition (~32% cell viability or ~68% growth inhibition) at 100 μM (Figure 5B). Similarly, 2-NBDG (~24% inhibition by flow), but not 2DG uptake inhibition (no inhibition) (Table 1), matched with growth inhibition by #12 in MCF-7 cells (~31% growth inhibition) at 50 μM (Figure 5C). As for WZB117 at 50 μM, ~73% glucose uptake inhibition determined by 2-NBDG uptake (~27% 2-NBDG uptake by flow cytometry), ~98% glucose uptake inhibition determined by 2-DG uptake (Table 1) and ~60% growth inhibition determined by the MTT assay (Figure 5C) were observed in MCF-7 cells. Thus, in our hands, although 2DG uptake determined by indirect measurement provided a wider detection window, glucose uptake inhibition determined directly by measuring 2-NBDG transport seems to correlate better with the growth inhibitory effect in both SKOV3 and MCF-7 cells. The degrees of correlation between GLUT1 inhibition and growth inhibitory activity varied in different cell lines that may in part be due to differences in GLUT1 expression levels and/or genetic characteristics. However, it remains a possibility that other off-target effects may contribute to the growth inhibitory effect of these GLUT1 inhibitors. Furthermore, although potential GLUT1 inhibitors #12, #16, #43 and #69 were selected in silico based on the crystal structure of hGLUT1 and validated using cell lines primarily expressing GLUT1, whether these compounds can also inhibit other GLUTs and whether the inhibitory activities toward other GLUTs also contribute to the growth inhibitory effect remain to be determined.

Although cancer cells exhibit increased glycolysis and depend more on this pathway for ATP generation, inhibition of glycolysis alone may not be sufficient to kill the malignant cells [2]. Thus, proper combination of multiple target-specific agents may be required to effectively kill cancer cells. Metformin, a widely used antidiabetic agent, was shown to synergize with 2DG against prostate cancer cells [22] and ovarian cancer cells [25]. The combination of metformin and STF31, a known GLUT1 inhibitor, also synergistically inhibited the growth of triple-negative breast cancer [30] and ovarian cancer cells [31]. In this study, we found a synergistic anticancer effect between a novel GLUT1 inhibitor #12 and metformin in SKOV3 cells, which express primarily the GLUT1 isoform of GLUTs [21]. The combination of WZB117 and metformin also showed a synergistic effect, suggesting that inhibition of GLUT1 at least in part contributed to the synergism between #12 and metformin in SKOV3 cells (Figure 6 and Appendix A).

Compound #43 was also a promising GLUT1 inhibitor based on several different assays and displayed the best growth inhibitory effect in the low concentration range in MCF-7 cells. Interestingly, #43 has a similar structure as caffeine, which has been reported to inhibit glucose uptake by binding to the nucleotide-binding site of GLUT1 [32]. We have recently reported that #43 can enhance the anticancer activity of cisplatin in both estrogen receptor-positive and triple-negative breast cancer cells via enhancing the DNA damaging effect and modulating the Akt/mTOR and MAPK signaling pathways, which is at least in part associated with GLUT1 inhibition [33].

In summary, we identified four potential GLUT1 inhibitors #12, #16, #43 and #69 from the NCI chemical library with diversified pharmacophores by a high-throughput non-radioactive cell-based method using 2-NBDG. These results were further validated using 2-NBDG uptake by flow cytometry and 2DG uptake assay coupling with enzymatic reactions for indirect measurement of 2DG. Different glucose uptake assays gave rise to somewhat different results, and inhibition of glucose uptake may not necessarily correlate well with inhibition of cancer cell growth. Nevertheless, we found that #12, showing the best inhibition of 2-NBDG uptake and growth in SKOV3 cells, also exhibited a synergistic anticancer effect against SKOV3 cells in combination with metformin. Furthermore, the combination of #12 and metformin also significantly suppressed cell migration, indicating a potential role in preventing cancer invasion.

## 4. Materials and Methods

### 4.1. Cell Lines and Cell Culture 

SKOV3 cells were cultured in McCoy’s 5A medium supplemented with 10% FBS, 1.5 mM l-glutamine and antibiotics including 100 units/mL of penicillin, 100 µg/mL of streptomycin and 0.25 µg/mL of amphotericin B. COS-7 and HepG2 cells were grown in low-glucose, and MCF-7 cells were grown in high-glucose Dulbecco’s Modified Eagle’s Medium supplemented with 10% FBS, 2 mM l-glutamine and antibiotics. Cells were cultured at 37 °C in a humidified 5% CO_2_ atmosphere.

### 4.2. RT-PCR

Total RNA was extracted using the TRIzol^TM^ Reagent followed by reverse transcription using the Magic RT cDNA synthesis kit (Bio-Genesis Technologies, Taipei, Taiwan). PCR was then conducted using primer pairs for *hGLUT1* (product size: 113 bp) and *hGLUT2* (product size: 159 bp). The HepG2 sample was included as a positive control for *hGLUT2*. A housekeeping gene *hGAPDH* was used as a positive control for RT-PCR (product size: 226 bp). PCR primers and PCR conditions are shown in Appendix A. The PCR products were analyzed using 4% agarose gel electrophoresis.

### 4.3. 2-NBDG Uptake Assay

Cells were seeded in 96-well plates (3–5 × 10^4^ cells/well), allowed to reach 100% confluence and subjected to the glucose uptake assay. After washing with Kreb’s Ringer Bicarbonate (KRB), buffer containing 129 mM NaCl, 4.7 mM KCl, 1.2 mM KH_2_PO_4_, 1.2 mM MgSO_4_, 2.0 mM CaCl_2_, 5.0 mM NaHCO_3_ and 10 mM HEPES, pH 7.4 [34], cells were incubated with test compounds and 200 μM of 2-NBDG in KRB buffer for the indicated time at 37 °C. Cells were then washed twice with cold KRB buffer, and examined under the microscope before being lysed in a buffer containing 1% sodium deoxycholate, 1% NP-40, 40 mM KCl and 20 mM Tris-HCl, pH 7.5 [35], and an aliquot of each lysate was transferred to a black 96-well plate for the detection of 2-NBDG fluorescence at an excitation wavelength of 475 nm and an emission wavelength of 550 nm [20]. Cells incubated with KRB buffer without 2-NBDG was included for the subtraction of background autofluorescence.

### 4.4. 2-NBDG Uptake Assay by Flow Cytometry

Cells were plated in 12-well plates (2.5 × 10^5^ cells/well) overnight, rinsed with PBS and pretreated with test compounds or vehicle control in PBS or KRB for 1 h at 37 °C and then 2-NBDG was added to each well (200 µM) for 30 min or they were treated with test compounds along with 200 µM 2-NBDG for 1.5 h. Cells were harvested by trypsinization, resuspended in cold PBS and then subjected to flow cytometric analysis as described [20].

### 4.5. Glucose Uptake-Glo^TM^ Assay

Cells were seeded in 96-well plates (1–5 × 10^4^ cells/well) overnight and subjected to the Glucose Uptake-Glo™ assay according to the manufacturer’s instructions (Promega, Madison, WI, USA). Briefly, cells were washed with KRB buffer and incubated for 10 min with test compounds or vehicle control in KRB buffer at 37 °C. Then, 2DG was added to a final concentration of 1 mM for 10 min at room temperature. Stop buffer and neutralization buffer were added sequentially to stop the reaction, and finally, detection reagent was added to each well for 1 h at room temperature. An aliquot of each lysate was then transferred to a white 96-well plate and the luminescence was detected using a GloMax^®^ 96 Microplate Luminometer (Promega, Madison, WI, USA).

### 4.6. Cell Viability Assay and Combination Index Analysis

Cells were seeded in 96-well plates (2–5 × 10^3^ cells/well) overnight and then treated with test compounds or the DMSO vehicle control for 72 h and subjected to the MTT assay. The absorbance was measured at 570 nm with 690 nm as a reference wavelength using a SpectraMax Paradigm microplate reader. IC_50_ values were calculated using GraphPad Prism 6 (GraphPad Software Inc., La Jolla, CA, USA). The combination index (CI) values were calculated by CompuSyn software. CI values < 1, =1 and >1 refer to synergistic, additive and antagonistic effects, respectively [36].

### 4.7. Colony Formation Assay

Cells were seeded in 6-well plates (1000 cells/well) and treated with indicated drugs for 10 days. Colonies were stained with 0.25% crystal violet in 20% methanol and photographed. Colonies with at least 50 cells were counted [37].

### 4.8. Apoptosis Assay by Annexin V-FITC/PI Double Staining

Cells harvested by trypsinization after drug treatment were resuspended in 1× Annexin V-FITC binding buffer, stained using the Annexin V Apoptosis Detection kit (Santa Cruz Biotechnology, Santa Cruz, CA) according to the manufacturer’s instructions and analyzed by flow cytometry.

### 4.9. Wound Healing Cell Migration Assay 

SKOV3 cells were seeded in 24-well plates and allowed to reach 100% confluence. A wound was then scratched, followed by drug treatment. Images were photographed at 0 and 24 h. The migration capacity was quantified by ImageJ 1.49v software with MRI_Wound_Healing_Tool.

### 4.10. Data Analysis 

Data are presented as mean ± SEM of the indicated independent experiments. Statistical analysis was performed using one-way ANOVA for comparison of multiple groups, and two-tailed Student’s *t*-test for comparison of two groups. *P*-values less than 0.05 were considered statistically significant.

## Figures and Tables

**Figure 1 molecules-27-08106-f001:**
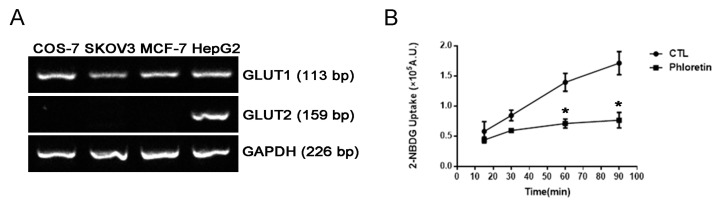
Validation of the conditions of the high-throughput 2-NBDG uptake assay. (**A**) GLUT1 and GLUT2 mRNA expression in COS-7, SKOV3 and MCF-7 cells determined by RT-PCR. RT-PCR products were analyzed by agarose gel electrophoresis, and the HepG2 sample was included as a positive control for GLUT2. (**B**) Time–course study of the 2-NBDG uptake assay in COS-7 cells. Inhibition of 2-NBDG uptake by 50 μM phloretin was determined after 15, 30, 60 or 90 min of incubation. Data are presented as mean ± SEM of three independent experiments except phloretin at 15 min (two independent experiments). Statistical significance was assessed by two-tailed Student’s *t*-test compared with the vehicle control. *, *p* < 0.05.

**Figure 2 molecules-27-08106-f002:**
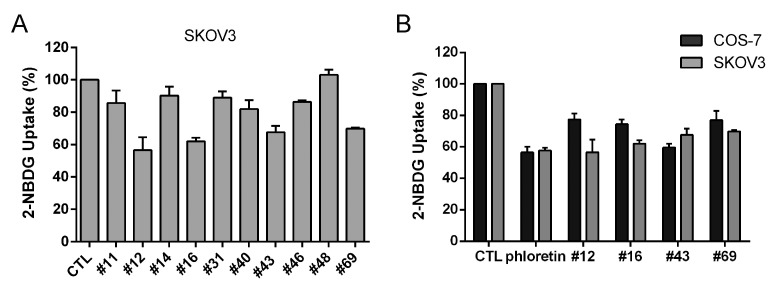
Inhibition of glucose uptake by potential GLUT1 inhibitors. (**A**) 2-NBDG uptake in SKOV3 cells of top 10 hits obtained from the assay in COS-7 cells. The concentration of compounds was 100 μM except #14 and #31 (50 μM). (**B**) Comparison of 2-NBDG uptake results in COS-7 and SKOV3 cells. The concentration of #12, #16, #43 and #69 was 100 μM, and 50 μM phloretin was included as a positive control. Data are presented as mean ± SEM of at least three independent experiments.

**Figure 3 molecules-27-08106-f003:**
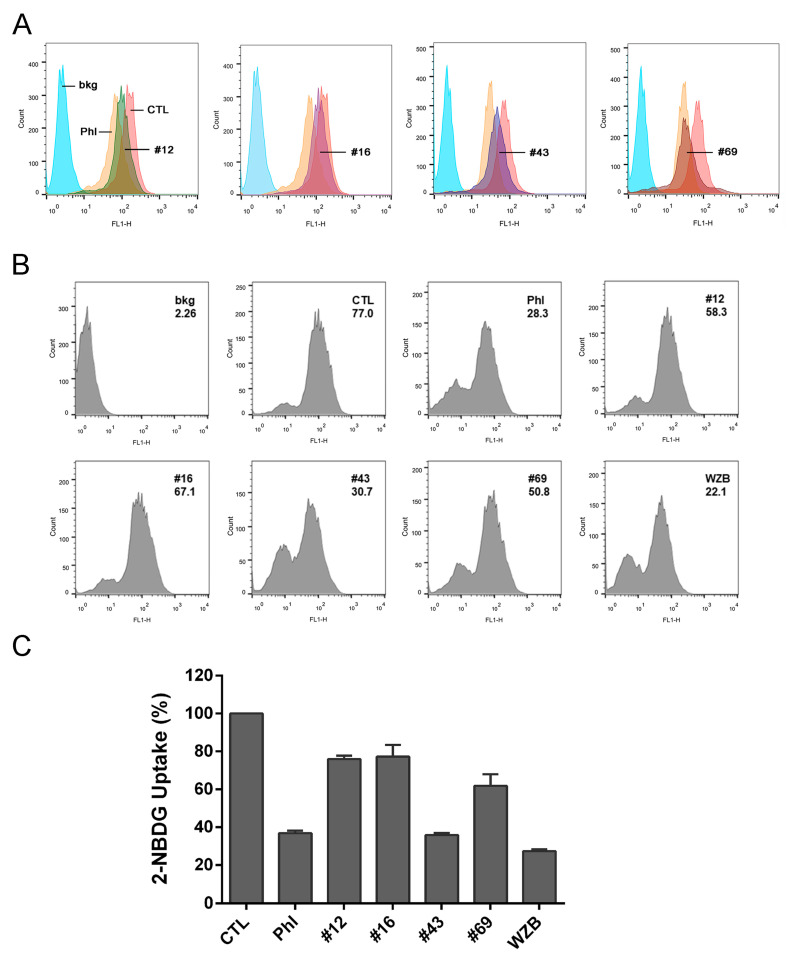
2-NBDG uptake assay by flow cytometry of potential GLUT1 inhibitors in SKOV3 and MCF-7 cells. (**A**) SKOV3. Cells were pretreated with 100 μM of NCI compounds or 50 μM phloretin (Phl) for 1 h and then incubated further in the presence of 200 μM 2-NBDG for 30 min and harvested for analysis. bkg: cell background, CTL: vehicle control. (**B**) MCF-7. Cells were treated with 50 μM of NCI compounds, Phl or WZB117 (WZB) along with 200 μM 2-NBDG for 1.5 h and harvested for analysis. A set of histograms is shown and geomean of each sample is indicated. (**C**) Quantitative data of MCF-7 cells. Cell background was subtracted from the geomean and 2-NBDG uptake was calculated relative to the vehicle control (100%). Data are presented as mean ± SEM of three independent experiments.

**Figure 4 molecules-27-08106-f004:**
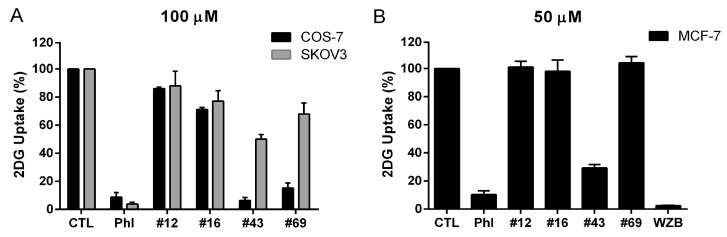
Glucose Uptake-Glo^TM^ assay of potential GLUT1 inhibitors in COS-7, SKOV3 and MCF-7 cells. (**A**) Inhibition of 2DG uptake by 100 μM of NCI hits in COS-7 and SKOV3 cells. Phloretin (Phl) was included as a positive control. (**B**) Inhibition of 2DG uptake by 50 μM of NCI hits, Phl and WZB117 (WZB) in MCF-7 cells. The percentage of 2DG uptake was calculated relative to the untreated vehicle control (CTL). Data are presented as mean ± SEM of at least three independent experiments except WZB (*n* = 2).

**Figure 5 molecules-27-08106-f005:**
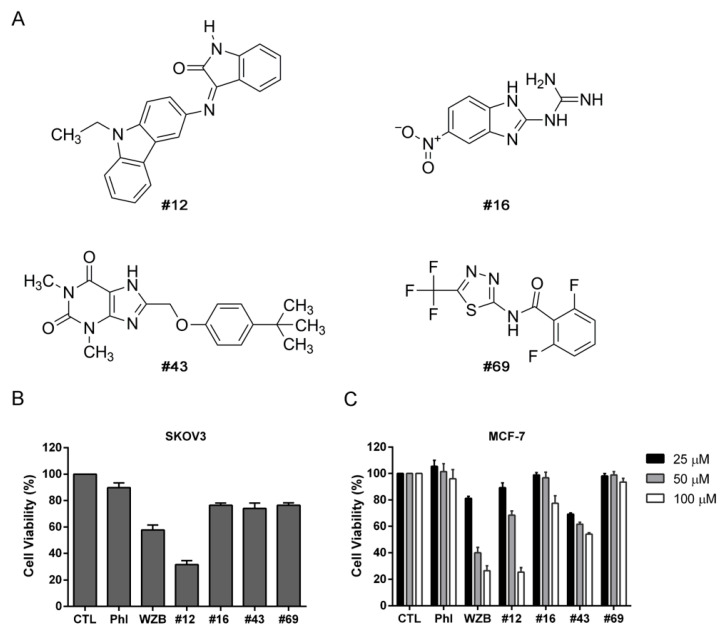
Chemical structures and growth inhibitory effects of potential GLUT1 inhibitors in SKOV3 and MCF-7 cells. (**A**) Chemical structures of NCI compounds #12, #16, #43 and #69. (**B**) Growth inhibitory effect of 100 μM of NCI hits in SKOV3 cells. (**C**) Growth inhibitory effect of 25, 50 and 100 μM of NCI hits in MCF-7 cells. Phloretin (Phl) and WZB117 (WZB) were included for comparison. Cells were treated with compounds for 72 h, followed by the MTT assay. Cell viability was calculated relative to the untreated vehicle control (CTL). Data are presented as mean ± SEM of at least three independent experiments.

**Figure 6 molecules-27-08106-f006:**
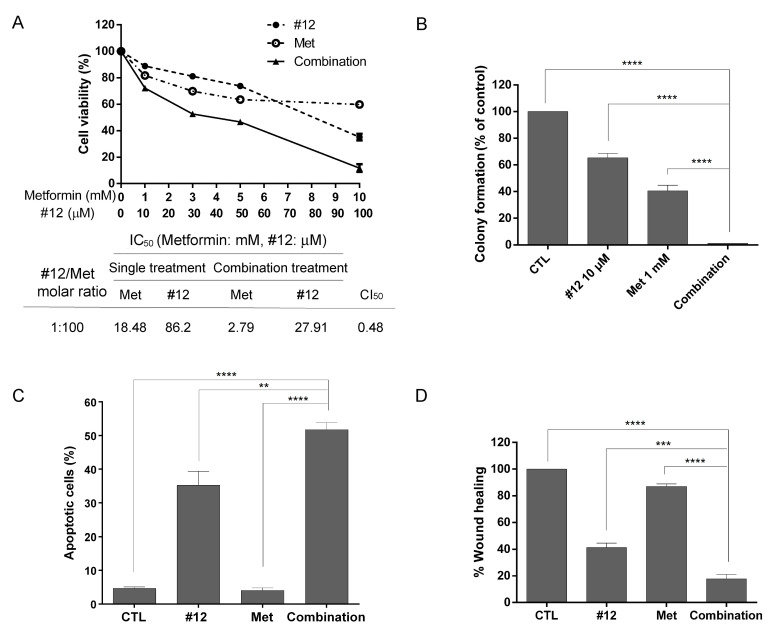
#12 synergistically enhances the anticancer effect of metformin in SKOV3 cells. (**A**) #12 synergized with metformin. Cell viability was measured by the MTT assay after 72 h of drug treatment. (**B**) Colony formation assay of the combination of 10 μM #12 and 1 mM metformin. (**C**) The combination of #12 and metformin effectively induced apoptosis. SKOV3 cells were treated with 100 μM #12 and/or 10 mM metformin for 72 h and then subjected to Annexin V-FITC/PI double staining and flow cytometric analysis. (**D**) The combination of 100 μM #12 and 10 mM metformin significantly inhibited cell migration determined by wound healing assay 24 h after drug treatment. CTL: vehicle control; Met: metformin. Data are presented as mean ± SEM of at least three independent experiments. Statistical significance was assessed by one-way analysis of variance (ANOVA). **, *p* < 0.01; ***, *p* < 0.001, ****, *p* < 0.0001.

**Table 1 molecules-27-08106-t001:** Comparison of glucose uptake results obtained from different methods in COS-7, SKOV3 and MCF-7 cells.

**% Uptake in COS-7 ^a^**	**2-NBDG**	**2DG**
**Phl**	53.83 ± 0.87	8.72 ± 3.22
**#12**	77.46 ± 3.82	86.04 ± 1.06
**#16**	74.36 ± 2.99	71.27 ± 1.40
**#43**	59.68 ± 2.28	6.38 ± 2.36
**#69**	76.96 ± 5.91	15.24 ± 3.46
**% Uptake in SKOV3 ^a^**	**2-NBDG**	**2DG**
**Phl**	55.58 ± 2.87	3.81 ± 1.35
**#12**	56.46 ± 8.05	88.06 ± 10.40
**#16**	62.07 ± 2.17	77.11 ± 7.40
**#43**	67.66 ± 3.93	50.03 ± 3.34
**#69**	69.87 ± 0.72	67.96 ± 7.82
**% Uptake in MCF-7 ^b^**	**2-NBDG flow**	**2DG**
**Phl**	36.87 ± 1.31	10.32 ± 2.76
**#12**	75.96 ± 1.75	101.07 ± 4.16
**#16**	77.28 ± 6.21	98.07 ± 8.12
**#43**	35.84 ± 1.11	29.24 ± 2.52
**#69**	61.81 ± 6.14	104.18 ± 4.41
**WZB**	27.35 ± 0.98	2.43 ± 0.25

^a^ The concentration of compounds was 100 μM. ^b^ The concentration of compounds was 50 μM.

## Data Availability

Not applicable.

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
