# Peer review of "Discovery of New Glucose Uptake Inhibitors as Potential Anticancer Agents by Non-Radioactive Cell-Based Assays"

_molecules, 2022, doi:10.3390/molecules27228106_

Round 1
Reviewer 1 Report
Authors propose in this paper to explore the inhibition of GLUT1 transporter by four selected inhibitors from a library. Inhibition of glucose uptake were evaluated using glucose analogues directedly by flow cytometry by competition with a florescent 2-NBDG uptake or indirectly 2-deoxy-D-glucose uptake in three cells lines : SKOV3, COS-7 or MCF-7. It’s important to underlie the key role of metabolic inhibitors in fighting against cancer cells and this paper aims at exploring this strategy.
Several points need to be clarified :
1- Figure 1: At the mRNA level, MCF-7 express GLUT1 but not GLUT2. However, MCF-7 express also GLUT6 mRNA (see protein Atlas database). This could be considered in RT-PCR experiment. Did the 3 cell lines express the same level of GLUT protein isoforms?
2- Figure 2B: the controls are missing in the comparison of 2-NBDG uptake for both cell lines.
3- Figure 3C: how do you explain the difference in the level of inhibition (#12 and #43) when data are compared those in Figure 3C graph.
Table 1: again, how the authors explain the deep difference in the uptake of the two glucose analogues in different cells lines?
Author Response
We thank the reviewer for the positive comments and for bringing up some important issues and suggestions. The response to the reviewer’s comments and suggestions are as the following:
Reviewer #1
Authors propose in this paper to explore the inhibition of GLUT1 transporter by four selected inhibitors from a library. Inhibition of glucose uptake were evaluated using glucose analogues directedly by flow cytometry by competition with a florescent 2-NBDG uptake or indirectly 2-deoxy-D-glucose uptake in three cells lines : SKOV3, COS-7 or MCF-7. It’s important to underlie the key role of metabolic inhibitors in fighting against cancer cells and this paper aims at exploring this strategy.
Several points need to be clarified:
1-Figure 1: At the mRNA level, MCF-7 express GLUT1 but not GLUT2. However, MCF-7 express also GLUT6 mRNA (see protein Atlas database). This could be considered in RT-PCR experiment. Did the 3 cell lines express the same level of GLUT protein isoforms?
Ans: We thank the reviewer for the comment. Indeed, there are many GLUT isoforms and we have added more detailed information of GLUT isoforms to the introduction:
“GLUTs belong to a family of transmembrane proteins with 14 members divided into three classes based on sequence similarity. Class I GLUTs include GLUT1-4 and GLUT14, class II GLUTs include GLUT5, GLUT7, GLUT9 and GLUT11, and class III GLUTs include GLUT6, GLUT8, GLUT10, GLUT12 and HMIT. These GLUT isoforms have different substrate specificity, transport capability and tissue distribution, and play important roles regulating glucose uptake and metabolism. Among them, GLUT1-4 have been well studied. Normally, GLUT1 is expressed in erythrocytes and blood brain barrier of the brain, responsible for basal glucose uptake. GLUT2 is mainly expressed in liver, small intestine, kidney and pancreas for transport of glucose and fructose, and glucose sensing. GLUT3 is expressed in neurons. GLUT4 is expressed in muscle, heart and adipose tissue responsible for insulin-stimulated glucose transport. Tumor cells usually overexpress GLUTs, primarily GLUT1; therefore, GLUT1 could be a potential target for anticancer therapy [4,5].”
Indeed, the three cell lines used in our studies may express multiple GLUT isoforms. Although this is a critical issue, it is difficult to determine the expression of all the isoforms. Therefore, we focused on GLUT1 which is considered the major isoform overexpressed in cancers including breast cancer and ovarian cancer. As shown in Figure 1A, these cell lines expressed GLUT1 mRNA with small variations. We are aware of this issue regarding isoforms and have described it in section 2.1 and Discussion:
“It remains to be determined whether COS-7 and SKOV3 cells express other GLUT isoforms. Nevertheless, it has been reported that GLUT1 but not GLUT2, 3 or 4 is overexpressed in human ovarian cancer [21]. Thus, compounds that can inhibit glucose uptake in COS-7 and SKOV3 could be potential GLUT1 inhibitors.” (section 2.1)
“The degrees of correlation between GLUT1 inhibition and growth inhibitory activity varied in different cell lines that may in part due to differences in GLUT1 expression levels and/or genetic characteristics. However, it remains a possibility that other off-target effects may contribute to the growth inhibitory effect of these GLUT1 inhibitors. Furthermore, although potential GLUT1 inhibitors #12, #16, #43 and #69 were selected in silico based on the crystal structure of hGLUT1 and validated using cell lines primarily expressing GLUT1, whether these compounds can also inhibit other GLUTs and whether the inhibitory activities toward other GLUTs also contribute to growth inhibitory effect remain to be determined.” (the third paragraph of Discussion)
2-Figure 2B: the controls are missing in the comparison of 2-NBDG uptake for both cell lines.
Ans: We thank the reviewer for bringing up this issue and have added the controls to Figure 2B.
3-Figure 3C: how do you explain the difference in the level of inhibition (#12 and #43) when data are compared those in Figure 3C graph.
Table 1: again, how the authors explain the deep difference in the uptake of the two glucose analogues in different cells lines?
We do not have a definite explanation for the differences in the inhibitory activity of #12 and #43 in different cell lines or using 2-NBDG or 2DG. Possible explanations are revised in section 2.4:
“These discrepancies may be in part due to differences in the detection of glucose uptake (direct vs. indirect detection), incubation time (30-90 min vs. 10 min), concentrations of substrates used (200 μM 2-NBDG vs. 1 mM 2DG) or GLUT1 expression levels. Further investigation is required to clarify this issue.”
Reviewer 2 Report
The authors present a nice work in screening GLUT1 inhibitors as potential anti-cancer agents. Four hit compounds were identified using 2-NBDG based high-throughput screening and validated by flow cytometry and Glucose Uptake-Glo assay. Anticancer activity studies suggested synergistic anticancer activity from #12 and metformin. By comparing the glucose uptake results measured by 2-NBDG and 2DG, the authors found that glucose uptake inhibitory ability measured by 2-NBDG correlated better with anticancer activity. This is informative for researchers in related field. The manuscript can be accepted after addressing some minor issues:
1. Please provide the chemical structures of known GLUT1 inhibitors such as phloretin, resveratrol, cytochalasin B, WZB117, BAY-876, which were mentioned in the introduction section.
2. Section 2.3, please give a brief introduction on the principle of Glucose Uptake-Glo assay.
3. Please explain why a molar ratio of 1:100 was chosen in studying the synergistic anticancer activity of #12 and metformin.
Author Response
We thank the reviewer for the positive comments and suggestions. The response to the reviewer’s comments and suggestions are as the following:
Reviewer #2
The authors present a nice work in screening GLUT1 inhibitors as potential anti-cancer agents. Four hit compounds were identified using 2-NBDG based high-throughput screening and validated by flow cytometry and Glucose Uptake-Glo assay. Anticancer activity studies suggested synergistic anticancer activity from #12 and metformin. By comparing the glucose uptake results measured by 2-NBDG and 2DG, the authors found that glucose uptake inhibitory ability measured by 2-NBDG correlated better with anticancer activity. This is informative for researchers in related field. The manuscript can be accepted after addressing some minor issues:
- Please provide the chemical structures of known GLUT1 inhibitors such as phloretin, resveratrol, cytochalasin B, WZB117, BAY-876, which were mentioned in the introduction section.
Ans: We have included the chemical structures of known GLUT1 inhibitors such as phloretin, resveratrol, cytochalasin B, WZB117, BAY-876 in Supplementary Materials Figure S1.
- Section 2.3, please give a brief introduction on the principle of Glucose Uptake-Glo assay.
Ans: We have included the principle of Glucose Uptake-Glo assay in section 2.3:
“Cells were seeded in 96-wells and pretreated with compounds for 10 min at 37oC before 2DG was added for further incubation for 10 min at room temperature. 2DG is converted to 2DG 6-phosphate (2DG6P) by hexokinase once transported into the cell. 2DG6P, not metabolized further and trapped inside the cell, can be quantified by coupling to an enzymatic detection system to generate luminescence. The detection reagent contains glucose 6-phosphate dehydrogenase (G6PDH), NADP+, reductase, luciferase and proluciferin. 2DG6P can be oxidized to 6-phosphodeoxygluconate, and NADP+ is simultaneously reduced to NADPH by G6PDH. NADPH is then used by reductase to convert proluciferin to luciferin which is used by luciferase to generate a luminescent signal that is proportional to the concentration of 2DG6P.”
- Please explain why a molar ratio of 1:100 was chosen in studying the synergistic anticancer activity of #12 and metformin.
Asn: Since the molar ratio of #12 and metformin IC50 values was ~1:200 (86.3 mM:18.48 mM), we tested 1:50, 1:100 and 1:200 molar ratios of the combination and found that a molar ratio of 1:100 had the best synergistic anticancer activity. Therefore, this ratio was chosen for further studies. We have included this information in section 2.6:
“The molar ratio of #12 and metformin IC50 values in SKOV3 cells was approximately 1:200, the 1:50, 1:100 and 1:200 molar ratios were then tested in a preliminary study, and 1:100 exhibiting the best combinatorial effect was chosen for further studies.”